# Learned pseudo-random number generator: WGAN-GP for generating statistically robust random numbers

Kiyoshiro Okada[1,3], Katsuhiro Endo[1], Kenji Yasuoka[1], Shuichi Kurabayashi[2,3]*

**1** Department of Mechanical Engineering, Keio University, Yokohama, Japan, **2** Graduate School of Media and Governance, Keio University, Fujisawa, Kanagawa, Japan, **3** Cygames Research, Cygames Inc., Shibuya, Tokyo, Japan

* kurabayashi_shuichi@cygames.co.jp

**Data Availability Statement:** All relevant data are within the paper.

**Funding:** The authors received no specific funding for this work.

## Abstract

Pseudo-random number generators (PRNGs) are software algorithms generating a sequence of numbers approximating the properties of random numbers. They are critical components in many information systems that require unpredictable and nonarbitrary behaviors, such as parameter configuration in machine learning, gaming, cryptography, and simulation. A PRNG is commonly validated through a statistical test suite, such as NIST SP 800-22rev1a (NIST test suite), to evaluate its robustness and the randomness of the numbers. In this paper, we propose a Wasserstein distance-based generative adversarial network (WGAN) approach to generating PRNGs that fully satisfy the NIST test suite. In this approach, the existing Mersenne Twister (MT) PRNG is learned without implementing any mathematical programming code. We remove the dropout layers from the conventional WGAN network to learn random numbers distributed in the entire feature space because the nearly infinite amount of data can suppress the overfitting problems that occur without dropout layers. We conduct experimental studies to evaluate our learned pseudo-random number generator (LPRNG) by adopting cosine-function-based numbers with poor random number properties according to the NIST test suite as seed numbers. The experimental results show that our LPRNG successfully converted the sequence of seed numbers to random numbers that fully satisfy the NIST test suite. This study opens the way for the "democratization" of PRNGs through the end-to-end learning of conventional PRNGs, which means that PRNGs can be generated without deep mathematical know-how. Such tailor-made PRNGs will effectively enhance the unpredictability and nonarbitrariness of a wide range of information systems, even if the seed numbers can be revealed by reverse engineering. The experimental results also show that overfitting was observed after about 450,000 trials of learning, suggesting that there is an upper limit to the number of learning counts for a fixed-size neural network, even when learning with unlimited data.

**Competing interests:** The authors have declared that no competing interests exist.

## Introduction

With the increasingly unpredictable and nonarbitrary behaviors required by information systems in various fields, random numbers have played a crucial role in implementing unpredictable and dynamic behaviors [1]. For example, cryptography [2, 3], gaming [4], machine learning [5, 6], and a wide range of simulations such as molecular simulation [7–11] and phase field simulation [12, 13] have utilized random numbers to implement unpredictable and nonarbitrary behaviors. Random numbers are a fundamental tool for implementing fairness and an essential software component for implementation. There are two common tools used to generate random numbers: true random number generators (TRNGs) utilizing physical phenomena and pseudo-random number generators (PRNGs) implemented as software algorithms. TRNGs adopt physical phenomena with randomnesses, such as temporal properties of operating system user processes, thermal noise, shot noise, electronics noise, and the emission timing of radioactive decay, to generate random numbers. Many researchers have developed well-known TRNG implementations, starting with low speed rates up to 300 Gb/s random bit generation (RBG) [14–16], and high speed rates up to 2 Tb/s RBG [17–28], and the fastest one of 250 Tb/s RBG was developed by Kim et al. [29]. Such TRNGs are often utilized in high-risk domains where genuine unpredictability is required, including security and finance. The major disadvantage of TRNGs is the long time required to generate many random numbers compared with PRNGs, which is due to their dependence on physical phenomena and the need for specific hardware. PRNGs have been developed as general-purpose software modules for many years and have been widely adopted in domains where a vast array of random numbers are required in information systems, such as the security field. PRNGs are algorithms that rapidly generate uncorrelated and random sequences of numbers that appear to be sufficiently complex and random for ordinary purposes. A PRNG is validated through a statistical test suite, such as NIST SP 800-22rev1a (NIST test suite) [30, 31], which is commonly used to evaluate the robustness and fairness of the generated random numbers and consists of 15 statistical tests. PRNG algorithms, such as Mersenne Twister (MT) [32] and Xorshift [33], and cryptographic PRNGs are deterministic, producing the same sequence of numbers with a unique period when the same initial values, called "seeds", are input.

Because many PRNG algorithms are open and publicly available, applying PRNGs to online applications and services requires extensive care to hide the seeds and implementation code, thus preventing malicious users from predicting the next number to be generated. It is straightforward for well-trained engineers to predict the periodic behavior of a PRNG if they know the seed number and the algorithm through reverse engineering. Although many application developers adopt the encryption of seeds and code obfuscation as good practice when utilizing a PRNG, it is difficult to completely hide implementation details in modern smartphone applications. There is an urgent need to invent a new random number generation model to allow application software developers without advanced mathematical skills to create tailor-made PRNGs whose behavior is publicly unpredictable and fully satisfies the NIST test suite.

Machine learning (ML) technologies can be used to realize artificial intelligence for studying algorithms and statistical models that allow computer systems to perform tasks without explicit instructions. These ML technologies have been considered beneficial for creating new PRNGs. The statistical models currently widely used are a type of function approximator called neural networks. With the successful adoption of deep neural networks (DNNs) with multiple hidden layers in many fields [34], many researchers have attempted to use them to create PRNGs by approximating the mathematical functions of existing PRNGs. For example, the Elman neural network [35], recurrent layers network [36, 37], recurrent neural networks with

long short-term memory units (LSTMs) [38, 39], and Hopfield neural network [40–42] are well-known recurrent neural network models that have been applied to generate PRNGs. Over the last decade, PRNG methods using new deep learning models, such as reinforcement learning [39, 43] and generative adversarial networks (GANs) [44, 45], have emerged and attracted the attention of researchers.

Given the availability of many ML technologies, a recent key question has been how to create a new PRNG by "end-to-end" learning [44], where ML technologies directly learn the behavior of the existing PRNGs without any preprocesses or post-processes. End-to-end deep learning can replace a complex ML system that involves multiple stages of processing from the initial data input to the final result output with a single large neural network with multiple layers and modules that perform various processes. Such end-to-end and fully automatic generation of PRNGs will enable ordinary application developers to create their own PRNGs customized to their applications and systems. As examples of end-to-end neural network PRNGs, De Bernardi et al. [44] and Oak et al. [45] have applied GANs [46] to learn the behavior of existing PRNGs that allow end-to-end random number generation. However, these studies [44, 45] did not adopt the recommended NIST test suite configurations involving 1,000 test runs (e.g., 10 test runs in the previous studies). To our knowledge, there are no ML-based or DNN-based PRNGs that fully satisfy the NIST test suite under the recommended evaluation criteria. A novel method to realize a learned random number generator that fully satisfies the officially recommended settings of the NIST test suite is desired to achieve end-to-end learning for PRNGs having production-level credibility.

Thus, we propose a new end-to-end random number generation model rather than just brushing up on previous research. We call it as the learned PRNG (LPRNG), which adopts a GAN with the Wasserstein distance (WGAN [47]), which is already known as a sophisticated version of GAN. This model learns from inexhaustible random numbers obtained from MT [32], a well-known PRNG algorithm, and generates practical random numbers that pass all the tests in the NIST test suite under the NIST recommended settings, including over 1,000 test runs. To our knowledge, this is the first automatically generated PRNG having the sufficient quality to be deployed in production, owing to it fully satisfying the NIST test suite. Our key modification of the WGAN is to remove the dropout layers, which markedly improved the generation of random numbers. Fig 1 shows a schematic overview of this study. To verify the robustness of this approach, we conducted evaluations that adopt cosine function-based seed values with poor random number properties to avoid information leakage in the learning phase. We also confirmed that these seed values have poor randomness using the NIST test suite. We sequentially observed the learning processes of the LPRNG by which its capability to transform an input seed into a random number with better randomness is improved as the learning progresses up to 900,000 iterations. The experimental results show that our model can learn "randomness" from the nearly infinite amount of training data without early overfitting problems, despite our model not including dropout layers. Interestingly, we found that overfitting occurs after about 450,000 iterations even with an infinite amount of training data, which indicates the existence of an upper limit to the number of learning counts for a fixed-size neural network.

Our proposed method using only end-to-end deep learning can democratize the random number generator, enabling ordinary engineers to develop their own PRNGs according to their needs. Our proposed method enables random number generation algorithms to be developed without deep mathematical knowledge or experience, which are traditionally considered indispensable, making it easy to create new and disposable random number generation algorithms. This means that a new publicly unknown and unpredictable random number generation algorithm can be used with confidence. In addition, the neural networks used in our

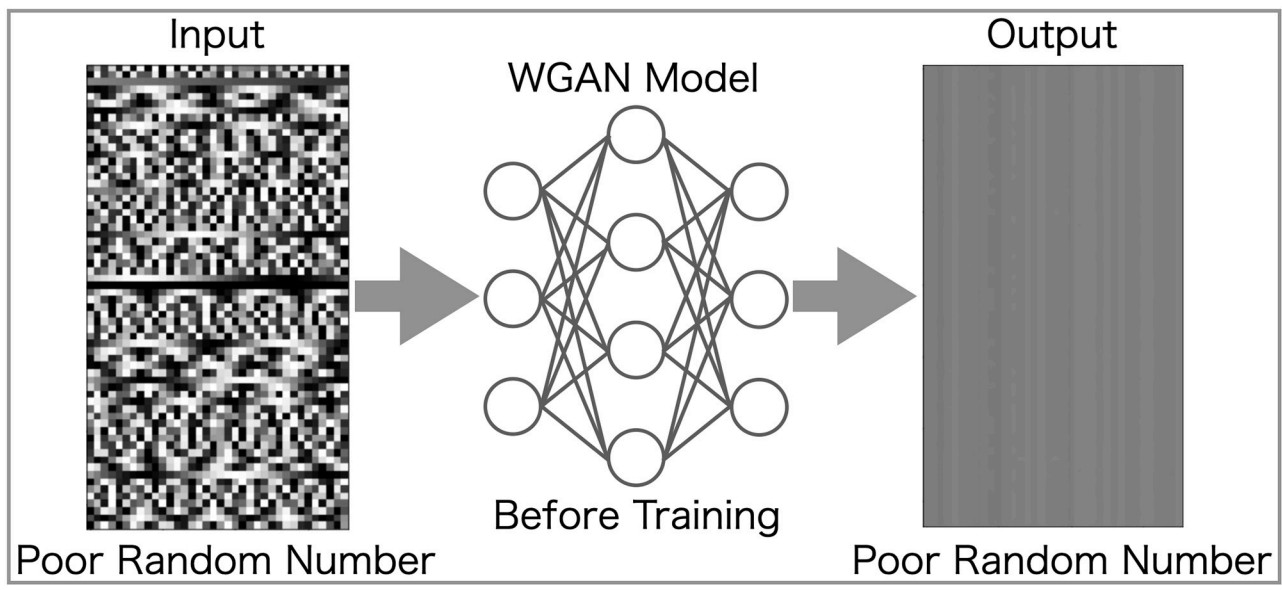

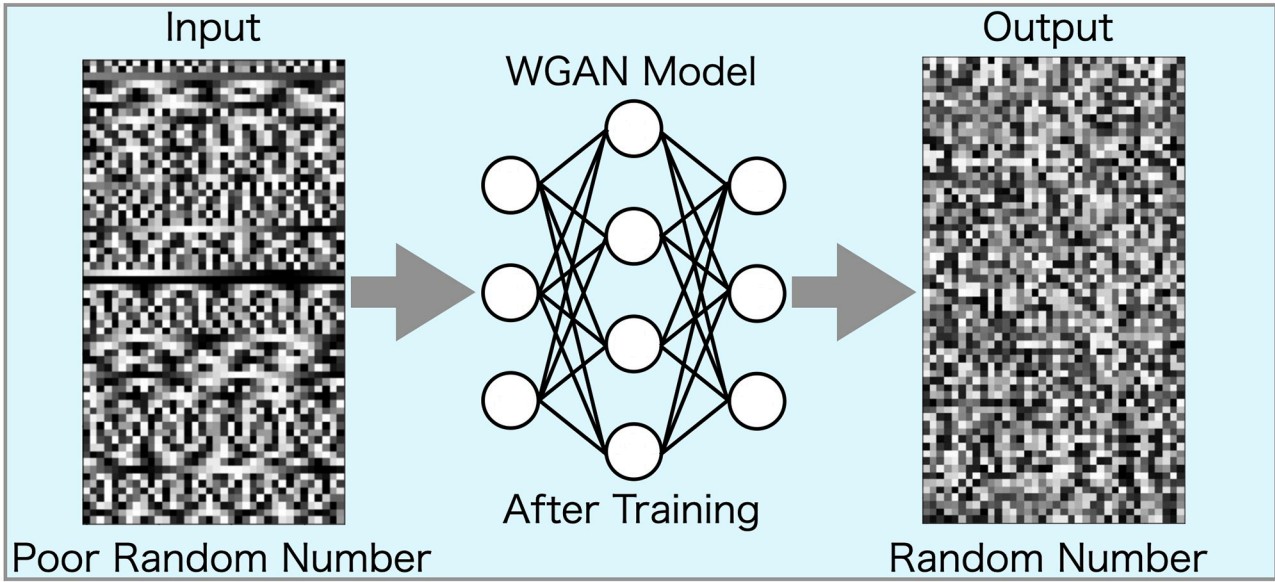

**Fig 1. Schematic of this research.** Upper image: WGAN model before training, which generates random numbers with poor properties. Lower image: WGAN model after training, which converts poor random numbers into random numbers with good properties.

method are both portable and robust against reverse engineering. Here, portability means that they are independent of the computer architecture and can be operated on any computer, and robustness against reverse engineering means that the algorithms in the models are difficult to interpret by humans. DNNs are black-box approximators. Since our LPRNG use an algorithm

given by black-box approximators, they provide no human insights into the inner functioning of the PRNGs.

Our NN-based random number generator has two benefits: reverse engineering resistance and cross-platform portability among servers and clients. To clarify these benefits, we will first outline how commonly PRNGs are used in mobile applications such as games and social networking services. Pseudo-random numbers are widely used not only in high-risk areas like encryption and security but also in areas that require randomness and non-arbitrariness, such as the parameter for natural and smooth animation in user interfaces and the control parameters for computer-operated Non-Player Characters (NPCs) in games. PRNGs are particularly acknowledged as an essential technology in the gaming industry, which has a significant impact on determining the winner of a game. Yet, the number of behaviors that violate game regulations, such as reverse engineering [48] to the seed of a PRNG to anticipate the next random number, has increased exponentially due to the generalization of game development tools such as Unity [49]. It is crucial to provide a simple and portable method to prevent the reverse engineering of a PRNG that is used carelessly.

Thus, game industries have adopted encrypting random number generation processes and frequently changing the decryption key, approximately every few weeks, as the simplest and most practical way to prevent reverse engineering. Such a brute-force method is effective so long as the frequency of key changes exceeds the time and cost necessary to decrypt or to find the decryption key. However, due to the fact that the algorithm for the pseudo-random number generator is already known and limited in number such as Mersenne twister [32], y Multiply-With-Carry (MWC) [50], Xorshift [33], and The 64-bit Maximally Equidistributed $F_2$-Linear Generators with Mersenne Prime Period (MELG-64) [51], once a seed location in memory is decrypted, the next random number generation result may be simply anticipated. As new pseudorandom number generation algorithms are developed only every few years at least, it is totally impractical to try to develop a new PRNG algorithm every week to prevent guessing the random number generators.

Our Learned PRNG allows the random number generator to be automatically created weekly by ordinary engineers who are not experts in mathematics. This renders it impossible to predict the behavior of applications utilizing Learned PRNGs, even if the memory location of the seed is disclosed, because it is difficult to investigate the behavior of the learned PRNG using commonly available reverse engineering techniques. The Learned PRNG offers a novel counter-reverse-engineering strategy that enhances the update frequency of the random number generation period, which can be used with the usual approach that often updates the encryption key. Our Learned PRNG model allows us to generate a new PRNG periodically every week from a data set of random number sequences generated by existing methods. Our research will drastically reduce the cost of delivering a new PRNG to edge devices such as smartphones, allowing us to update PRNGs more frequently than it would cost to crack and identify the PRNG and its surrounding systems.

As for the portability of the Learned PRNG model, it is a "machine/language neutral" data structure that is independent of a particular programming language or CPU architecture. The services of today must be compatible with multiple platforms, such as Apple's iOS, Google's Android, Microsoft's Windows, Apple's macOS, and Linux. In addition, we employ multiple CPUs, including x86-64, ARM, and RISV-V, on both the client-side and server-side. Numerous programming languages, such as C/C+, Python, Java, JavaScript, C#, and Swift, are used to implement services. Consequently, the cost of adapting a new PRNG to all platforms, CPUs, and programming languages is exceptionally high. Our Learned PRNG model, on the other hand, can be represented as open neural network exchange (ONNX) [52], a machine-neutral and language-neutral model that is independent of these platforms, operating systems, and

programming languages. Dedicated runtimes [53] and compilers [54] enable our Learned PRNG model to effectively infer random numbers on a variety of platforms and hardware. Smartphones of the present day are typically equipped with specialized chips for the rapid execution of neural networks, such as the Apple Neural Engine, which can efficiently execute NN models. Therefore, the delivery of Learned PRNGs to edge devices is practical.

The proposed random number generation method using ML has the following applications:

1. Our LPRNG method realizes the one-off creation of new statistically valid PRNGs. Such one-off PRNG prevents mathematical or software-engineering experts from exploiting their knowledge by taking unfair profit from PRNG. Our LPRNG provides unpredictable behavior for everyone, including ordinary people and experts, because it includes no human insights into the inner functioning of the PRNG. This feature makes it impossible for anyone to use corruptly optimized and tuned random numbers to achieve desirable results.

2. Our LPRNG provides high-level anti-tampers for software developers who use PRNGs on the client-side such as PCs and smartphones. It is very difficult for malicious users to predict the next random numbers generated by our LPRNG, even if the seed number might be disclosed through the reverse-engineering because users cannot know the learning process and parameters essential to investigate the behavior of LPRNGs.

In the following, we show how to create an input seed, and we discuss the process by which our ML model outputs random numbers with sufficient properties in terms that pass statistical tests.

A significant difficulty with the Learning PRNG is that it requires a seed sequence equal to the quantity of random numbers to be generated. Most known PRNG algorithms use an asymptotic formula to produce random numbers constantly by recursively using the output as input to generate subsequent random numbers. Further research will validate that the Learned PRNG passes the NIST test suite even when the output of WGAN is utilized as input to WGAN to create continuous random numbers.

## Methods

### Wasserstein Generative Adversarial Network (WGAN)

Goodfellow et al. [46] initially proposed Generative adversarial networks (GANs), which are a type of deep generative model, and they are commonly used for generating images and predicting simulation results [55].

In this algorithm, a generator network $G$ competes with a discriminator network $D$. After learning process, the generator will generate samples $x$ that deceive the discriminator whereas the discriminator will correctly discriminate whether the samples are generated by the generator or not.

$$V(D, G) = \mathbb{E}_{x \sim p_r}[\log D(x)] + \mathbb{E}_{x \sim p_g}[\log(1 - D(x))]. \tag{1}$$

Where $D(x)$ is the probability that the discriminator decides a sample $x$ is a real sample ($x \sim p_g$: probability distribution of false samples, $x \sim p_r$: probability distribution of real samples). GAN learning follows a mini-max game, represented by

$$\min_G \max_D V(D, G). \tag{2}$$

Where $V(D, G)$ is the objective function of the GAN.

Conventional GANs often pose significant challenges in their training process, primarily attributable to the instability of the learning processes [56, 57]. Two of these challenges stand out: one is the vanishing gradient problem, a well-known issue where the gradient of the loss function shrinks to an infinitesimally small value, inhibiting effective weight updates during backpropagation. The second, referred to as mode collapse, describes a phenomenon where the generator persistently yields identical, remarkably similar, or a restricted set of outputs, remaining unaffected by changes in the input noise vector. To address these predicaments that contribute to the learning instability, a variation of GANs known as Wasserstein GAN (WGAN) was proposed by Arjovsky et al. [47].

$W$ denotes the cost of transporting earth from one location to another. If the earth is a probability distribution function, $W$ is the minimum total work involved in converting one distribution to another. The GAN objective function $V(D, G)$ can be expressed using the Wasserstein distance as

$$
\begin{aligned}
V(D, G) &= W(p_r, p_g) \\
&= \max_{\phi} \mathbb{E}_{x \sim p_r}[D(x; \phi)] - \mathbb{E}_{z \sim p_z}[D(G(z; \theta); \phi)],
\end{aligned} \tag{3}
$$

where $D(x; \phi)$ is a function satisfying the Lipschitz continuity $\|D(x_1) - D(x_2)\| \leq \|x_1 - x_2\|$ and here, we used both spectral normalization [58] and gradient penalty [59] (Wasserstein GAN with gradient penalty: WGAN-GP).

## Training of Wasserstein Generative Adversarial Network

We built the machine learning model to predict the structure of the random numbers generated by MT using the WGAN [46, 47]. We trained the WGAN with the MT random numbers used as training data and used the WGAN as a model to output the random numbers. The WGAN trained with various random images is expected to generate images with many types of random structures.

Fig 2 shows the architecture of the WGAN. For each layer, the Leaky ReLU function is used as the activation function of the input and hidden layers. The output of the generator is scaled to the range of (0,1) using the sigmoid function. A schematic view of the training process is shown in Fig 3. Poor-quality random numbers generated by the cosine function (Eq 4) were used for the input seed of the generator. Input seed has 36 dimensions, and we used 64 batches. This input seed has low-quality randomness in terms of the NIST test results. Then, the discriminator decides whether the generated numbers are random numbers or low-quality random numbers. In this study, MT random numbers were used as training data. We set the dimension of the latent variable to 36 and defined each latent variable as a floating-point number in the range from 0 to 1 generated by the cosine function. The discriminator was trained five times for each training of the generator. One update of the parameters of the generator after updating the parameters of the discriminator five times was considered as one iteration. The RMSprop optimizer was used for the discriminator and generator. Spectral normalization [58] was implemented in all the linear layers in the discriminator. We also applied a gradient penalty [59].

## WGAN without dropout layers

In this study, we used the WGAN model, which excludes the dropout layers through tuning for random number generation by machine learning. We originally used a model with dropout layers, but the removal of the dropout layers markedly improved the quality of the generated random numbers. In general, as the structure of a neural network becomes increasingly

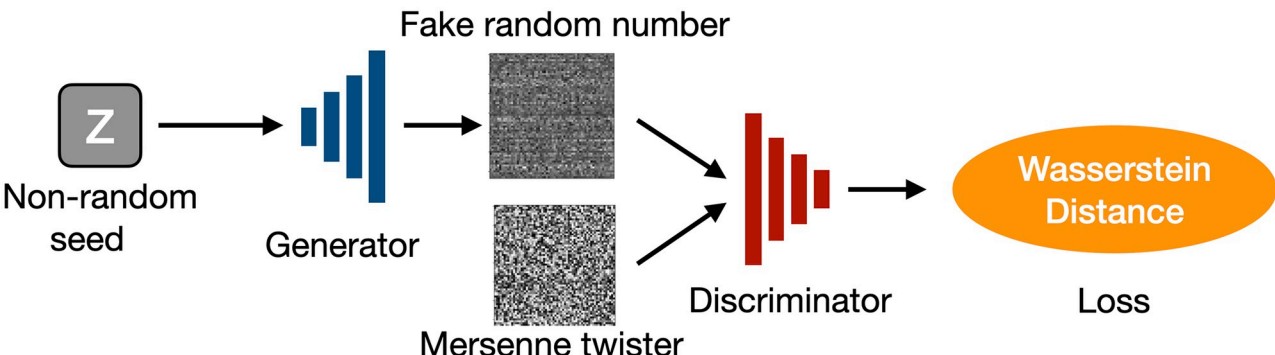

**Fig 2. WGAN model.** Upper figure: Generator model. Lower figure: discriminator model.

**Fig 3. Schematic representation of the learning process of the Wasserstein GAN.** At each iteration, the generator receives the latent variable $z$ and outputs a false image, and the discriminator receives the false and real images and calculates the loss using the Wasserstein distance.

complex, the weights of the neurons become optimized for the training data set. As it is, the model lacks generalization performance and becomes poor and can only be used for the training data set as if memorizing the data one by one. The percentage of correct answers in the training data will gradually increase as the learning proceeds, but the error in the test data will stop decreasing and start increasing again. Such a situation is called overfitting. The dropout layer is used to prevent this phenomenon. In our case, we have an infinite number of random numbers as the training data. Moreover, the learning process was accelerated by removing the dropout layers.

## NIST tests

To evaluate the randomness of the numbers generated from the machine learning model, we employed the NIST random number test suite, which consists of the following 15 tests, Frequency Test, Block Frequency Test, Cumulative Sums Test, Runs Test, Longest Run Test, Rank Test, FFT Test, Non-Overlapping Template Test, Overlapping Template Test, Universal Test, Approximate Entropy Test, Random Excursions Test, Random Excursions Variant Test, Serial Test, Linear Complexity Test. Please refer to the specification [30] for details.

We used each series of 1048576 bits for each test method, and the results of the 1000 series were used to make comprehensive results. As these tests were based on a statistical analysis of a random stream, a p-value was reported for each test or subtest, indicating the probability that the result is due to randomness. A significance level of 0.01 was used for all tests.

## The method of bit window extraction

Binary data must be prepared for evaluation using the NIST random number test. On the other hand, in machine learning, a decimal floating-point representation is generally used. In our experiments, we extracted the binary data from the floating-point representation, consisting of the sign, exponent, and mantissa parts. When considering a random number between 0 and 1, the sign part and the exponent part are almost unchanged. This means that the randomness of the bit sequence is low. On the other hand, some of the mantissa part may contain randomness. To use the randomness, we extracted several bits from the mantissa part that were found to have sufficient randomness in terms of NIST test results in the MT case, as shown in Fig 4. The results of the NIST tests are for these several bits from the mantissa part of the generated random numbers.

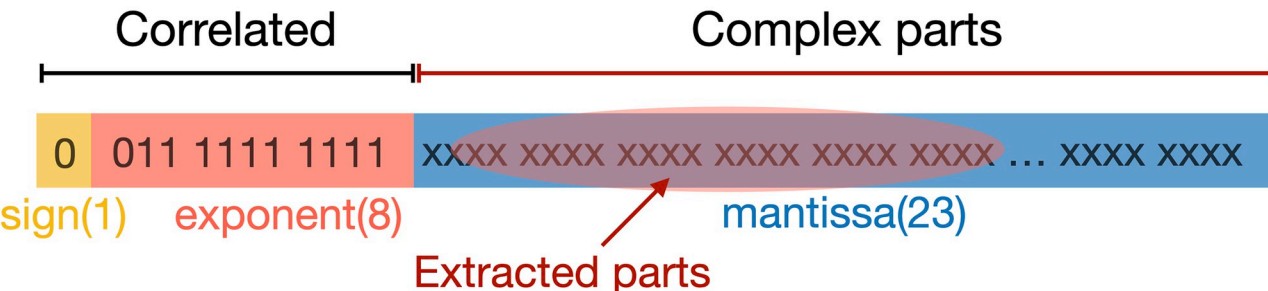

**Fig 4. Bit window extraction method.** We extracted several bits from the mantissa part that were found to have sufficient randomness in the case of MT-generated random numbers.

## Results and discussion

### Creating an input seed

The main objective of this study is to generate random numbers using machine learning techniques, specifically by using GANs. When attempting to generate random numbers using GANs, it is important to note that the input seed should not possess random properties. Although random numbers are commonly used as input seeds, in this study, using random numbers as input seeds would not allow us to determine whether the GAN learned the random properties to generate output or simply output the input seed's properties without learning any random properties. If a sequence of numbers without randomness is used as the input seed and the output has randomness through transformation by GAN, it can be considered that GAN has learned randomness and is able to transform it into a sequence of numbers with randomness. Therefore, we devised poor random input seeds that are suitable for GAN learning. There are many possibilities for poor random number sequences, but it is important to provide various numbers to the GAN during learning. we considered number sequences with regularity or periodicity as poor random number sequences and introduced trigonometric functions. Even if a sequence of numbers can be converted to random numbers in a GAN conversion, the relationship between batches of numbers cannot be controlled by the GAN. To address this issue, we used different initial phases ($R_2$) and periods ($R_1$) of trigonometric functions for each batch to reduce the similarity between number sequences in advance. This is also important in providing diverse input seeds for GAN learning. The finally selected input seed was represented by

$$\text{input} = \cos(2R_1\pi t + 2R_2\pi) \times \frac{1}{2} + 1, \tag{4}$$

where $0 \leq t < 36$ and $R_1$ and $R_2$ are random numbers in the range of $0 \leq t \leq 1$ which are introduced to reduce the similarity of the numerical sequences and to provide a variety of input seeds. The input was also adjusted to be between 0 and 1. A sequence of numbers generated by the cosine function with 64 unique periods is shown in Fig 5. Periodicity in the horizontal direction can be seen. The results of evaluating this numerical sequence by the NIST test in Table 1 show that this numerical sequence does not have sufficient randomness. This sequence of numbers was used as the initial input seed for machine learning. The use of these input seeds is also different from the previous studies.

### Transition of learning

Fig 6 shows the transition of the loss function. During the initial phase of training (approximately 250,000 iterations), it can be observed that the loss decreases as the learning progresses. However, in the latter half of the training, the loss starts to increase again. Fig 7(a) shows the NIST test results of the random numbers outputted at each learning step. We ran the NIST test after each set of 3000 iterations by connecting multiple batches of random numbers generated at each step. We divided the learning process into four regions denoted W, X, Y, and Z. In region W, the NIST score improved as the learning progressed. In region X, the NIST score fluctuated around 11, always failing to pass the Frequency, CumulativeSums, and Runs tests, often failing NonOverlappingTemplate test and rarely failing to pass the OverlappingTemplate, Universal, ApproximateEntropy, and Serial tests. In region Y, the generated random numbers finally passed the NIST test completely but often failed to pass the Frequency, CumulativeSums, Runs, and NonOverlappingTemplate tests. It was difficult for our model to pass the Frequency and CumulativeSums tests. Note that just by learning MT random numbers, it becomes possible to pass the NIST 15-item statistical test. Tables 2 and 3 respectively show the

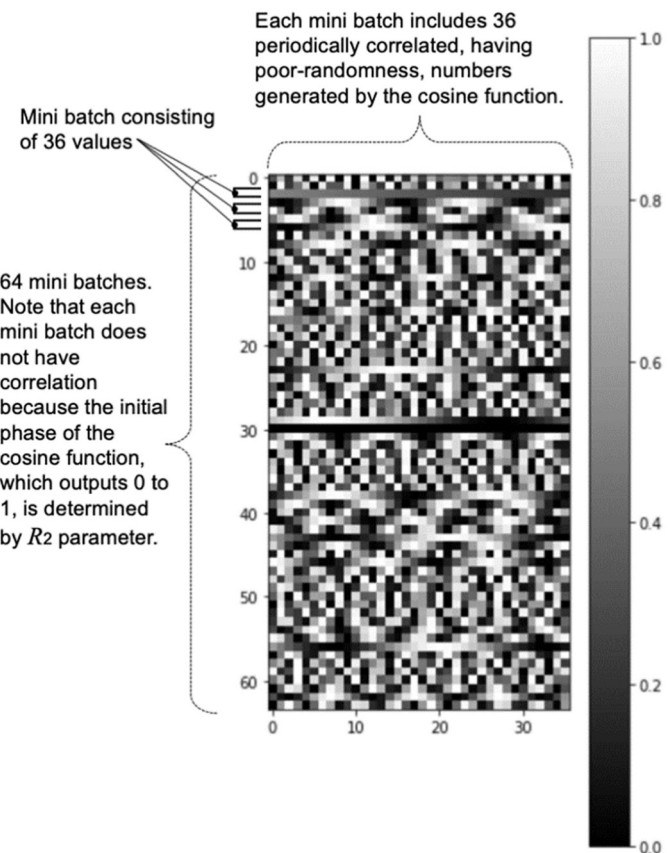

**Fig 5. Simple sequence of numbers generated by the cos function.** The horizontal axis represents a sequence of poor random numbers representing a single input seed, and the vertical axis represents the data from each of the 64 batches.

**Table 1. NIST statistical test results for input seed.**

| Statistical test | Pass rate | Uniformity of $p$-values | Pass/fail judgment |
|---|---|---|---|
| Frequency | 0.000 | Fail | Fail |
| BlockFrequency | 0.000 | Fail | Fail |
| CumulativeSums | 0.000 | Fail | Fail |
| CumulativeSums | 0.000 | Fail | Fail |
| Runs | 0.000 | Fail | Fail |
| LongestRun | 0.000 | Fail | Fail |
| Rank | 0.995 | Pass | Pass |
| FFT | 0.983 | Pass | Pass |
| NonOverlappingTemplate | 0.784 | 36/148 Pass | Fail |
| OverlappingTemplate | 0.000 | Fail | Fail |
| Universal | 0.541 | Fail | Fail |
| ApproximateEntropy | 0.000 | Fail | Fail |
| RandomExcursions | 0.000 | 0/8 Pass | Fail |
| RandomExcursionsVariant | 0.000 | 0/18 Pass | Fail |
| Serial | 0.000 | Fail | Fail |
| Serial | 0.800 | Fail | Fail |
| LinearComplexity | 0.992 | Pass | Pass |

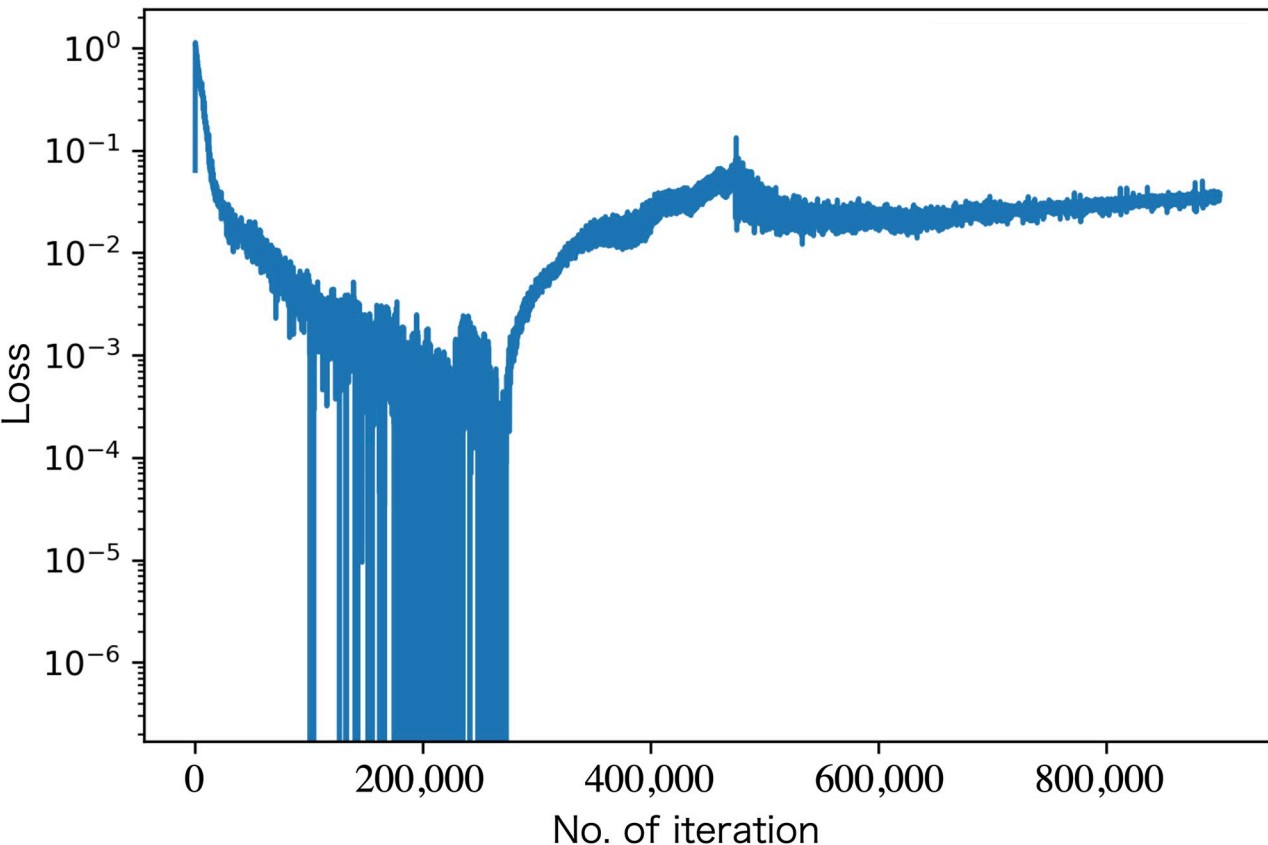

**Fig 6. Loss function as a function of a number of iterations.**

scores of the NIST tests for the random numbers outputted before and after learning in which a perfect score was obtained (the data for A and C in Fig 7). In images A, B, and C in Fig 7(b), the growth of randomness can be seen. Note that our model has difficulty in stably passing all NIST tests during continuous learning. However, we found that we can generate good random numbers even if our model learns to output a small amount of random numbers (but there are many batches), because the random numbers created by connecting multiple outputs consisting of 36 float type bitstreams pass the NIST test. Fig 8 shows the results when a drop-apt layer is added. As can be seen, the learning results are better when the Dropout layer is removed.

After achieving a perfect score on the NIST test in region Y, the score deteriorates in region Z. In this region, only the Rank, FFT, RandomExcursions, RandomExcursionsVariant, and LinearComplexity tests are usually passed, and ultimately the RandomExcursions and RandomExcursionsVariant tests are failed. Throughout the learning, Rank, FFT, and LinearComplexity tests are easily passed, whereas it is difficult to pass the Frequency and CumulativeSums tests. Fig 7(b) also shows that the random images after long-term learning (D) are not random, and the score of the NIST test is shown in Table 4. In this machine learning process, over fitting occurs even though an infinite random dataset is used. Over fitting is generally caused by the use of finite datasets, but our results show that it can also occur for infinite datasets. We believe that the early stopping of the algorithm is effective even when infinite datasets are used. We additionally discussed two cases where the random property of the input seed was better or worse.

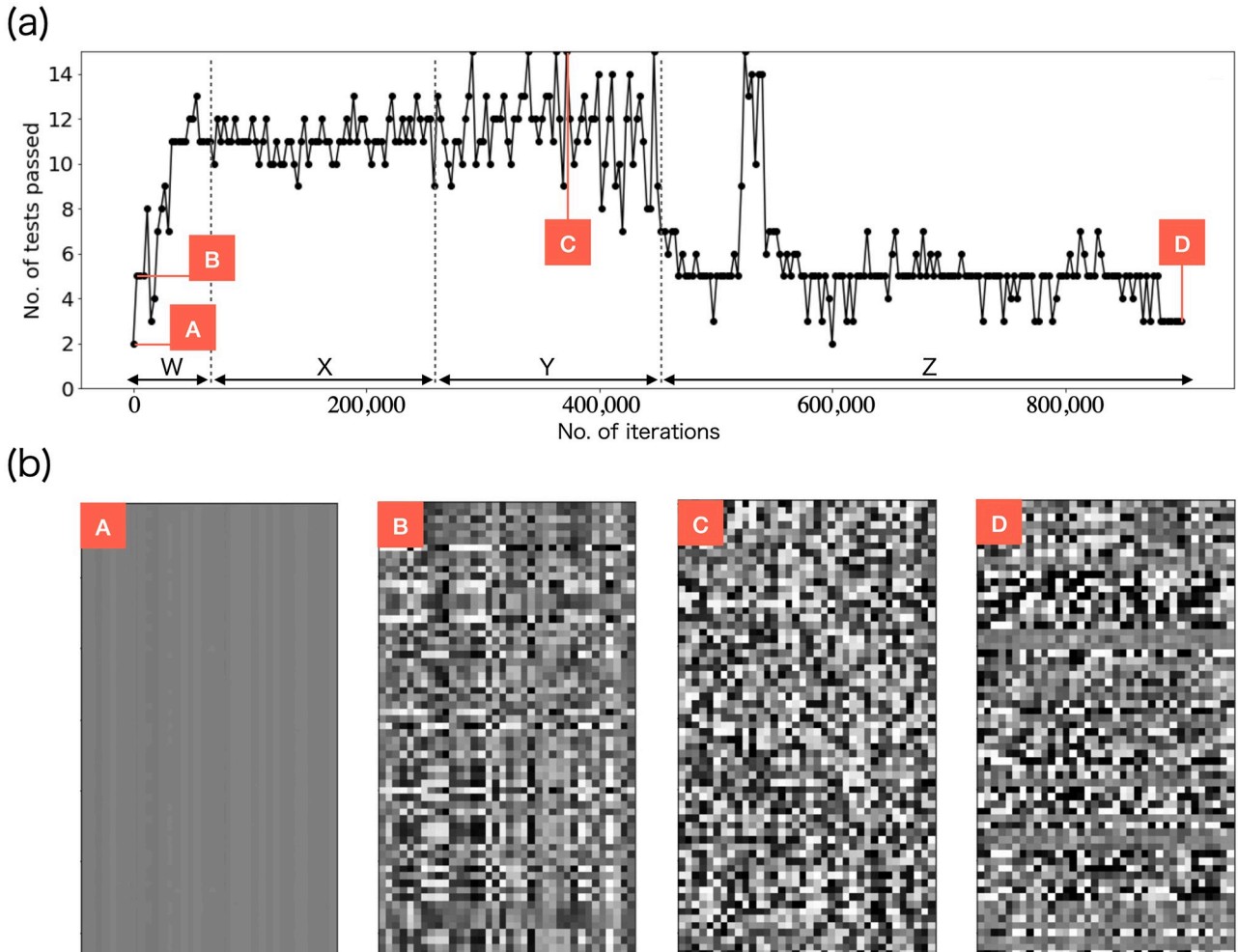

**Fig 7. NIST test results and images of generated random numbers at different numbers of iterations.** The NIST scores of the random numbers improve in the order A < B < C. The random number images show that the randomness increases. The random number performance degrades from C to D, indicating that over fitting occurs despite the use of infinite training data. One update of the parameters of the generator after updating the parameters of the discriminator five times was considered as one iteration.

1. What happens if we use input seeds that have random properties? In this case, the input seed will be the same as those widely used in general, which is more suitable for machine learning. Consequently, the output will also produce random numbers that clear the NIST tests. However, it is difficult to determine whether these random numbers are superior to those generated using input seeds made with trigonometric functions. This is because our study used the NIST test for random verification and we cannot compare the quality of the two outputs beyond the range of a perfect score. Therefore, in this case, it is also possible that the machine learning model does not learn the randomness property, but learns to output the property of the input seed as it is, making it difficult to determine whether the machine learning model has successfully imitated the random properties.

2. What happens if we use input seeds that have worse properties as random numbers? Although we developed input seeds using trigonometric functions, there are several methods to create input seeds with worse properties than these. For example, consider a

**Table 2. NIST statistical test results for data A in Fig 7 (before learning).**

| Statistical test | Pass rate | Uniformity of *p*-values | Pass/fail judgment |
|---|---|---|---|
| Frequency | 0.000 | Fail | Fail |
| BlockFrequency | 0.000 | Fail | Fail |
| CumulativeSums | 0.000 | Fail | Fail |
| CumulativeSums | 0.000 | Fail | Fail |
| Runs | 0.000 | Fail | Fail |
| LongestRun | 0.000 | Fail | Fail |
| Rank | 0.990 | Pass | Pass |
| FFT | 0.000 | Fail | Fail |
| NonOverlappingTemplate | 0.113 | 0/148 Pass | Fail |
| OverlappingTemplate | 0.000 | Fail | Fail |
| Universal | 0.000 | Fail | Fail |
| ApproximateEntropy | 0.000 | Fail | Fail |
| RandomExcursions | 0.000 | 0/8 Pass | Fail |
| RandomExcursionsVariant | 0.000 | 0/18 Pass | Fail |
| Serial | 0.000 | Fail | Fail |
| Serial | 0.000 | Fail | Fail |
| LinearComplexity | 0.987 | Pass | Pass |

numerical sequence where 1 always appears continuously. In this case, the value of the input seed becomes limited, and the learning process becomes extremely difficult, which results in the difficulty in obtaining random numbers with good properties as outputs. As one of our contributions to this study, we can emphasize that we have developed a way to create numerical sequences without random properties that do not significantly affect the machine learning process.

**Table 3. NIST statistical test results for data C in Fig 7 (successful area).**

| Statistical test | Pass rate | Uniformity of *p*-values | Pass/fail judgment |
|---|---|---|---|
| Frequency | 0.992 | Pass | Pass |
| BlockFrequency | 0.988 | Pass | Pass |
| CumulativeSums | 0.992 | Pass | Pass |
| CumulativeSums | 0.995 | Pass | Pass |
| Runs | 0.991 | Pass | Pass |
| LongestRun | 0.988 | Pass | Pass |
| Rank | 0.982 | Pass | Pass |
| FFT | 0.989 | Pass | Pass |
| NonOverlappingTemplate | 0.990 | 148/148 Pass | Pass |
| OverlappingTemplate | 0.989 | Pass | Pass |
| Universal | 0.991 | Pass | Pass |
| ApproximateEntropy | 0.987 | Pass | Pass |
| RandomExcursions | 0.990 | 8/8 Pass | Pass |
| RandomExcursionsVariant | 0.990 | 18/18 Pass | Pass |
| Serial | 0.988 | Pass | Pass |
| Serial | 0.987 | Pass | Pass |
| LinearComplexity | 0.991 | Pass | Pass |

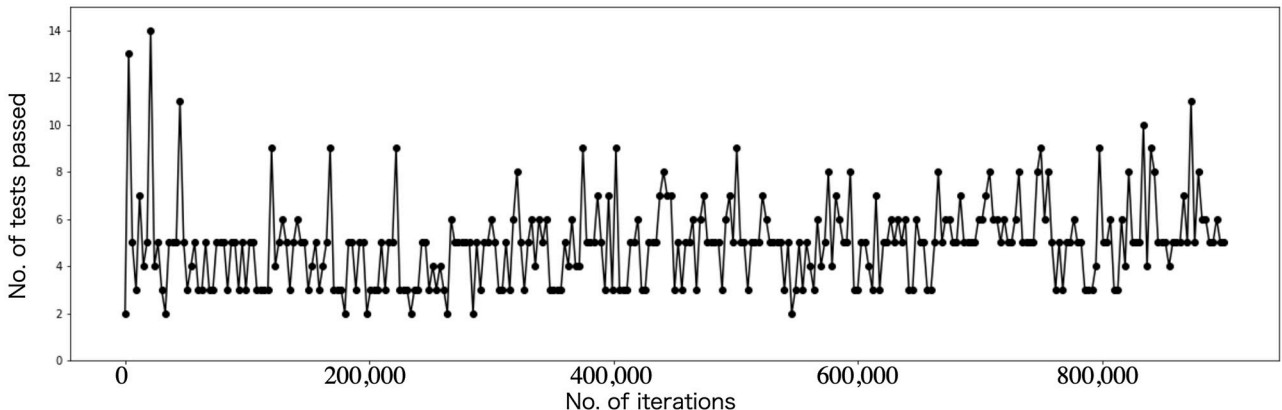

**Fig 8. NIST test results when WGAN learned with dropout layer.**

## Conclusion and future work

In this study, we succeeded in generating random numbers using a WGAN with poor random number seeds. The obtained random numbers were evaluated by the NIST random number tests and found to pass all the tests. We also observed over fitting even though infinite training data were used. Through training, our model was found to easily pass the Rank, FFT, and LinearComplexity tests, but not the Frequency and CumulativeSums tests. The mechanism of deep learning is still not fully understood. In this study, we clarified how the generator learns by examining the tests that were passed. The evaluation of learning by statistical tests, such as random number tests, may bring new insights into our understanding of deep learning. This research has various applications. 1. By using random numbers that follow an arbitrary distribution as the training data for learning, it may become possible to design a random number

**Table 4. NIST statistical test results for data D in Fig 7 (over fitting area).**

| Statistical test | Pass rate | Uniformity of *p*-values | Pass/fail judgment |
|---|---|---|---|
| Frequency | 0.000 | Fail | Fail |
| BlockFrequency | 0.000 | Fail | Fail |
| CumulativeSums | 0.000 | Fail | Fail |
| CumulativeSums | 0.000 | Fail | Fail |
| Runs | 0.000 | Fail | Fail |
| LongestRun | 0.000 | Fail | Fail |
| Rank | 0.992 | Pass | Pass |
| FFT | 0.984 | Pass | Pass |
| NonOverlappingTemplate | 0.900 | 75/148 Pass | Fail |
| OverlappingTemplate | 0.000 | Fail | Fail |
| Universal | 0.752 | Fail | Fail |
| ApproximateEntropy | 0.000 | Fail | Fail |
| RandomExcursions | 1.000 | 0/8 Pass | Fail |
| RandomExcursionsVariant | 1.000 | 0/18 Pass | Fail |
| Serial | 0.370 | Fail | Fail |
| Serial | 0.989 | Fail | Fail |
| LinearComplexity | 0.991 | Pass | Pass |

generation method that follows an arbitrary distribution. 2. Training machine learning to take computational efficiency into account will lead to the development of computationally efficient and fast random number generation methods. 3. Random number testing methods using machine learning have been investigated in previous studies [60]. The discriminator used in GAN training also can be used as a random number test as it distinguishes between random and poor random numbers. In addition, unlike the NIST test, there is no limit to the amount of input data, so a discriminator can be used to test random numbers for arbitrary input data sizes. Our results provide useful information for the use of machine learning and the future development of random number generation methods.

## Author Contributions

**Conceptualization:** Kiyoshiro Okada, Shuichi Kurabayashi.

**Data curation:** Kiyoshiro Okada.

**Formal analysis:** Kiyoshiro Okada.

**Investigation:** Kiyoshiro Okada, Shuichi Kurabayashi.

**Methodology:** Kiyoshiro Okada, Katsuhiro Endo, Kenji Yasuoka, Shuichi Kurabayashi.

**Project administration:** Shuichi Kurabayashi.

**Software:** Kiyoshiro Okada, Katsuhiro Endo.

**Supervision:** Katsuhiro Endo, Kenji Yasuoka, Shuichi Kurabayashi.

**Validation:** Kiyoshiro Okada, Katsuhiro Endo.

**Writing – original draft:** Kiyoshiro Okada.

**Writing – review & editing:** Kiyoshiro Okada, Kenji Yasuoka, Shuichi Kurabayashi.

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
