## [Decision Letter · Decision Letter 0]

11 Jan 2023

PONE-D-22-24211Learned Pseudo-Random Number Generator: WGAN-GP for Generating Statistically Robust Random NumbersPLOS ONE

Dear Dr. Kurabayashi,

Thank you for submitting your manuscript to PLOS ONE. After careful consideration, we feel that it has merit but does not fully meet PLOS ONE’s publication criteria as it currently stands. Therefore, we invite you to submit a revised version of the manuscript that addresses the points raised during the review process.Please address the comments of both reviewers carefully in order to proceed further.

We look forward to receiving your revised manuscript.

Kind regards,

Sheetal Kalyani

Academic Editor

PLOS ONE

Journal Requirements:

2. Thank you for submitting the above manuscript to PLOS ONE. During our internal evaluation of the manuscript, we found significant text overlap between your submission and previous work in the methods section.

We would like to make you aware that copying extracts from previous publications word-for-word is unacceptable. In addition, the reproduction of text from published reports has implications for the copyright that may apply to the publications.

Please revise the manuscript to rephrase the duplicated text, cite your sources, and provide details as to how the current manuscript advances on previous work. Please note that further consideration is dependent on the submission of a manuscript that addresses these concerns about the overlap in text with published work.

We will carefully review your manuscript upon resubmission and further consideration of the manuscript is dependent on the text overlap being addressed in full. Please ensure that your revision is thorough as failure to address the concerns to our satisfaction may result in your submission not being considered further.

Please note that PLOS ONE has specific guidelines on code sharing for submissions in which author-generated code underpins the findings in the manuscript. In these cases, all author-generated code must be made available without restrictions upon publication of the work. Please review our guidelines at https://journals.plos.org/plosone/s/materials-and-software-sharing#loc-sharing-code and ensure that your code is shared in a way that follows best practice and facilitates reproducibility and reuse.

Reviewers' comments:

Reviewer's Responses to Questions

**Comments to the Author**

1. Is the manuscript technically sound, and do the data support the conclusions?

Reviewer #1: Yes

Reviewer #2: Partly

2. Has the statistical analysis been performed appropriately and rigorously? 

Reviewer #1: Yes

Reviewer #2: Yes

3. Have the authors made all data underlying the findings in their manuscript fully available?

Reviewer #1: No

Reviewer #2: Yes

4. Is the manuscript presented in an intelligible fashion and written in standard English?

Reviewer #1: Yes

Reviewer #2: Yes

5. Review Comments to the Author

Reviewer #1: This paper shows an interesting application of WGAN to the generation of PRNs, using Mersenne Twister PRNG to produce training data. The authors show in Table 3 that the generator learns how to produce PRNs.

However, I was not able to find any reference to the source code. For this reason, as it is presented, this paper does not meet standard reproducibility criteria.

The source code used for the model and its training, together with the training data used for the experiments, should be provided. In this way the scientific community can reproduce and improve the experiments described in this paper.

As soon as the code is provided, I think that this paper describes a sound piece of research.

Reviewer #2: The authors propose using WGANs for PRNG. They use a cosine function to generate an input seed, which is poorly random. The output of the WGAN is said to produce random numbers which is validated using the NIST test suite. The method is interesting, however, no comparisons are provided with existing PRNG mechanisms. I have the following questions/ comments about the paper:

1. Please provide comparisons with existing methods that use GANs for PRNG, namely references [44] and [45]. The advantage of using WGAN (without dropout) over simple GANs is not quantified.

2. “The neural networks used in our method are both portable and robust against reverse engineering (difficult to interpret by humans)” (Lines 102-106). Please justify. Can a neural network be trained to reverse-engineer and learn the parameters of the WGAN?

3. The authors cite one of their key contributions as the removal of the dropout layers in WGAN (Line 82) to improve the quality of random numbers. The model still overfits after ~250,000 iterations according to Fig. 6. A numerical comparison with and without the dropout layers would be interesting.

4. While creating an input seed, how is the correlation removed beforehand? Also please justify how Eq (4) was chosen as the cosine function. Were other functions also tried in experiments?

5. This method uses a poorly random sequence as the seed. Does the output of the WGAN improve with the quality of randomness of the input seed?

6. In Line 291, the authors state that “It can be seen that the loss decreases as the learning progresses.” This statement is misleading as it suggests a steady decrease in loss with time when in Fig.6, loss clearly increases after ~250,000 iterations.

Minor comments:

1. In Eq(3), there is a typo: the symbol – is used instead of ~

2. The paper could be organised better, the Figures and Tables should ideally be placed close to the reference. For instance, Fig. 7 is referenced on Page 6 but is present only in Page 13.

3. Are the terms “non-random” and “poorly random” used interchangeably? In a few lines, the cosine function seed is referred to as non-random (Eg: Fig 3) and as poorly random in the Introduction.

4. In Line 194, the term “overlearning” is used. Overfitting is a more well-known term for that phenomenon.

6. PLOS authors have the option to publish the peer review history of their article (what does this mean?). If published, this will include your full peer review and any attached files.

Reviewer #1: No

Reviewer #2: No

---

## [Author Response · Author response to Decision Letter 0]

26 Feb 2023

For Reviewer #1

Thank you very much for your kind comments. Thanks to your suggestions, we were able to amend and meet standard reproducibility criteria. The source code developed in this study has been uploaded to the following GitHub link. We hope you will find it useful.

https://github.com/kiyoooooo/PlosOne_Learned_PRNG

We also write them in Data Availability Statement as below.

“All of our source codes developed in this study are freely available at https://github.com/kiyoooooo/PlosOne_Learned_PRNG. Every data including input seeds and output seeds can be reproduced by them.”

For Reviewer #2

Thank you very much for your kind comments. Thanks to your suggestions, we were able to amend and improve the text taking into consideration the questions and points you raised that we needed to pay attention to. Please find our response to the points and questions raised below.

Please provide comparisons with existing methods that use GANs for PRNG, namely references [44] and [45]. The advantage of using WGAN (without dropout) over simple GANs is not quantified.

This was not clearly explained at all in the previous version of our manuscript and we should have worded this better. We are sorry for the misleading presentation. 

In this study, we have redefined the problem and applied this machine learning architecture as a completely new approach, rather than brushing up on previous research. The GAN model used in our study is a widely known and applied model, and not a new AI model. After considering various versions of GAN models, we decided to use WGAN as a superior version of normal GAN. Although the theme of our study is similar to previous research, there are differences in the input seed, GAN architecture, and output method, making it difficult to make a strict comparison. However, according to the reports, while previous research utilized 10 test runs in the NIST test, we confirmed the results using 1000 test runs, which is standard practice. We recognize that one of the contributions of this research is the ability to generate random numbers using GANs based on the standard evaluation way of the NIST test.

Additional statement: 

e.g. 10 test runs in the previous studies (Line 69 on page 3 in the Introduction)

rather than just brushing up on previous research. (Line 75 on page 3 in the Introduction)

a GAN with the Wasserstein distance, which is already known as a sophisticated version of GAN. (Line 77 on page 3 in the Introduction)

The use of these input seeds is also different from the previous studies. (Line 337 on page 10 in the Results and Discussion)

“The neural networks used in our method are both portable and robust against reverse engineering (difficult to interpret by humans)” (Lines 102-106). Please justify. Can a neural network be trained to reverse-engineer and learn the parameters of the WGAN? 

Following your very helpful comments we could have updated the manuscript like below.

Additional statement: “Our NN-based random number generator has two benefits: reverse engineering resistance and cross-platform portability among servers and clients. To clarify these benefits, we will first outline how commonly PRNGs are used in mobile applications such as games and social networking services. Pseudo-random numbers are widely used not only in high-risk areas like encryption and security, but also in areas that require randomness and non-arbitrariness, such as the parameter for natural and smooth animation in user interfaces and the control parameters for computer-operated Non-Player Characters (NPCs) in games. PRNGs are particularly acknowledged as an essential technology of in the gaming industry, which have a significant impact on determining the winner of a game . Yet, the number of behaviors that violate game regulations, such as reverse engineering [1] to the seed of a PRNG to anticipate the next random number, has increased exponentially due to the generalization of game development tools such as Unity [2]. It is crucial to provide a simple and portable method to prevent the reverse engineering of a PRNG that is used carelessly.

Thus, game industries have adopted encrypting random number generation process and frequently changing the decryption key, approximately every few weeks, as the simplest and most practical way to prevent reverse engineering. Such a brute-force method is effective so long as the frequency of key changes exceeds the time and cost necessary to decrypt or to find the decryption key. However, due to the fact that the algorithm for the pseudo-random number generator is already known and limited in number such as Mersenne twister[3], MWC[4], Xorshift[5], and MELG-64[6], once a seed location in memory is decrypted, the next random number generation result may be simply anticipated. As new pseudorandom number generation algorithms are developed only every few years at least, it is totally impractical to try to develop a new PRNG algorithm every week to prevent to prevent guessing the random number generators.

Our Learned PRNG allows the random number generator to be automatically created weekly by ordinary engineers who are not expert of mathematics. This renders it impossible to predict the behavior of applications utilizing Learned PRNGs, even if the memory location of the seed is disclosed, because it is difficult to investigate the behavior of the learned PRNG using commonly available reverse engineering techniques. The Learned PRNG offers a novel counter-reverse-engineering strategy that enhances the update frequency of the random number generation period, which can be used with the usual approach that often updates the encryption key. Our Learned PRNG model allows us to generate a new PRNG periodically every week from a data set of random number sequences generated by existing methods. Our research will drastically reduce the cost of delivering a new PRNG to edge devices such as smartphones, allowing us to update PRNGs more frequently than it would cost to crack and identify the PRNG and its surrounding systems.

As for the portability of the Learned PRNG model, it is a "machine/language neutral" data structure that is independent of a particular programming language or CPU architecture. The services of today must be compatible with multiple platforms, such as Apple's iOS, Google's Android, Microsoft's Windows, Apple's macOS, and Linux. In addition, we employ multiple CPUs, including x86-64, ARM, and RISV-V, on both the client-side and server-side. Numerous programming languages, such as C/C+, Python, Java, JavaScript, C#, and Swift, are used to implement services. Consequently, the cost of adapting a new PRNG to all platforms, CPUs, and programming languages is exceptionally high. Our Learned PRNG model, on the other hand, can be represented as ONNX [7], a machine-neutral and language-neutral model that is independent of these platforms, operating systems, and programming languages. Dedicated runtimes [8] and compilers [9] enable our Learned PRNG model to effectively infer random numbers on a variety of platforms and hardware. Smartphones of the present day are typically equipped with specialized chips for the rapid execution of neural networks, such as the Apple Neural Engine, which can efficiently execute NN models. Therefore, the delivery of Learned PRNGs to edge devices is practical. 

A significant difficulty with the Learning PRNG is that it requires a seed sequence equal to the quantity of random numbers to be generated. Most known PRNG algorithms use an asymptotic formula to produce random numbers constantly by recursively using the output as input to generate subsequent random numbers. Further research will validate that the Learned PRNG passes the NIST test suite even when the output of WGAN is utilized as input to WGAN to create continuous random numbers.”

(On page 4 in the Introduction)

The authors cite one of their key contributions as the removal of the dropout layers in WGAN (Line 82) to improve the quality of random numbers. The model still overfits after ~250,000 iterations according to Fig. 6. A numerical comparison with and without the dropout layers would be interesting. 

Thank you for these questions as they raise important points we must consider for the wider achievements of our study. As you know, the Dropout layer is used to improve the generalization performance of machine learning models by avoiding overfitting for finite and biased data. In our research, we used infinite data and uniform random numbers, which are unbiased data, as the teacher data, so we removed the Dropout layer. However, thanks to your valuable suggestions, we have additionally validated and examined how the presence or absence of the Dropout layer affects the learning results.

Additional statement: Fig.8 shows the results when a drop-apt layer is added. As can be seen, the learning results are better when the Dropout layer is removed. (Line 397 on page 14 in the Results and Discussion, Fig. 8)

While creating an input seed, how is the correlation removed beforehand? Also please justify how Eq (4) was chosen as the cosine function. Were other functions also tried in experiments?

Thank you so much for bringing this to our attention. This was not clearly explained at all in the previous version of our manuscript and we should have worded this better. We added the explanations about how is the correlation removed and how Eq (4) was chosen as the cosine function like below. We were able to create the ideal input seed for training GANs using trigonometric functions and have not tested it with any other functions.

Original statement: 

“We experimented with ML using various input seed values with the aim of selecting input seeds that are non-random before the transformation by the ML model and random after the transformation. We found that sine and cosine functions with various periods for each batch generated good random numbers in the range of (0,1). Various periods were used for each batch because a GAN does not eliminate the batch-to-batch correlation, making it necessary to eliminate the correlations between batches beforehand.”

Corrected statement: 

“The main objective of this study is to generate random numbers using machine learning techniques, specifically by using GANs. When attempting to generate random numbers using GANs, it is important to note that the input seed should not possess random properties. Although random numbers are commonly used as input seeds, in this study, using random numbers as input seeds would not allow us to determine whether the GAN learned the random properties to generate output or simply output the input seed's properties without learning any random properties. If a sequence of numbers without randomness is used as the input seed and the output has randomness through transformation by GAN, it can be considered that GAN has learned randomness and is able to transform it into a sequence of numbers with randomness. Therefore, we devised non-random input seeds that are suitable for GAN learning. There are many possibilities for non-random number sequences, but it is important to provide various numbers to the GAN during learning. we considered number sequences with regularity or periodicity as non-random number sequences and introduced trigonometric functions. Even if a sequence of numbers can be converted to random numbers in a GAN conversion, the relationship between batches of numbers cannot be controlled by the GAN. To address this issue, we used different initial phases (R2) and periods (R1) of trigonometric functions for each batch to reduce the similarity between number sequences in advance. This is also important in providing diverse input seeds for GAN learning.”(on page 9 in the Results and Discussion)

To make Fig. 5 easier to understand, we have added explanations for each axis direction to the figure. (Fig.5)

Additional statement: “which are introduced to reduce the similarity of the numerical sequences and to provide a variety of input seeds” (Line 331 on page 10 in the Results and Discussion)

“The horizontal axis represents a sequence of non-random numbers representing a single input seed, and the vertical axis represents the data from each of the 64 batches.” (Fig.5)

This method uses a poorly random sequence as the seed. Does the output of the WGAN improve with the quality of randomness of the input seed?

Thank you for your insightful question. This is an important point that we should have made clear. The task we addressed in this study is to generate random numbers using machine learning. Therefore, we needed to investigate whether it is possible to convert input seeds without random properties to outputs with random properties. This is why we developed input seeds using trigonometric functions. In addition to this, based on the valuable questions we received, we divided this question into two separate cases:

Additional statement:

1. What happens if we use input seeds that have random properties?

In this case, the input seed will be the same as those widely used in general, which is more suitable for machine learning. Consequently, the output will also produce random numbers that clear the NIST tests. However, it is difficult to determine whether these random numbers are superior to those generated using input seeds made with trigonometric functions. This is because our study used the NIST test for random verification and we cannot compare the quality of the two outputs beyond the range of a perfect score. Therefore, in this case, it is also possible that the machine learning model does not learn the randomness property, but learns to output the property of the input seed as it is, making it difficult to determine whether the machine learning model has successfully imitated the random properties.

2. What happens if we use input seeds that have worse properties as random numbers?

Although we developed input seeds using trigonometric functions, there are several methods to create input seeds with worse properties than these. For example, consider a numerical sequence where 1 always appears continuously. In this case, the value of the input seed becomes limited, and the learning process becomes extremely difficult, which results in the difficulty in obtaining random numbers with good properties as outputs. As one of our contributions to this study, we can emphasize that we have developed a way to create numerical sequences without random properties that do not significantly affect the machine learning process.

(On page 13 in the Results and Discussion)

In Line 291, the authors state that “It can be seen that the loss decreases as the learning progresses.” This statement is misleading as it suggests a steady decrease in loss with time when in Fig.6, loss clearly increases after ~250,000 iterations.

Thank you so much for bringing this to our attention. We have corrected the text to fix this mistake like below.

Original statement: "It can be seen that the loss decreases as the learning progresses."

Corrected statement: "During the initial phase of training (approximately 250,000 iterations), it can be observed that the loss decreases as the learning progresses. However, in the latter half of the training, the loss starts to increase again.” (Line 342 on page 11 in the Results and Discussion)

Minor comments: 1. In Eq(3), there is a typo: the symbol – is used instead of ~ 2. The paper could be organised better, the Figures and Tables should ideally be placed close to the reference. For instance, Fig. 7 is referenced on Page 6 but is present only in Page 13. 3. Are the terms “non-random” and “poorly random” used interchangeably? In a few lines, the cosine function seed is referred to as non-random (Eg: Fig 3) and as poorly random in the Introduction. 4. In Line 194, the term “overlearning” is used. Overfitting is a more well-known term for that phenomenon.

(Line 224 on page 6 in the Methods)

We refrained from mentioning Fig. 7 in the first half of the article, and instead added an explanation in the caption of Fig. 7.

(Line 305 on page 9 in the Results and Discussion)

(Line 269 on page 7 in the Methods)

Thank you so much for bringing this to our attention. Based on your helpful comments, we were able to amend and improve the text taking into consideration the questions and points you raised that we needed to pay attention to.

---

## [Decision Letter · Decision Letter 1]

30 May 2023

Learned Pseudo-Random Number Generator: WGAN-GP for Generating Statistically Robust Random Numbers

PONE-D-22-24211R1

Dear Dr. Kurabayashi,

We’re pleased to inform you that your manuscript has been judged scientifically suitable for publication and will be formally accepted for publication once it meets all outstanding technical requirements.

Kind regards,

Sheetal Kalyani

Academic Editor

PLOS ONE

Additional Editor Comments (optional):

Reviewers' comments:

Reviewer's Responses to Questions

**Comments to the Author**

1. If the authors have adequately addressed your comments raised in a previous round of review and you feel that this manuscript is now acceptable for publication, you may indicate that here to bypass the “Comments to the Author” section, enter your conflict of interest statement in the “Confidential to Editor” section, and submit your "Accept" recommendation.

Reviewer #2: All comments have been addressed

2. Is the manuscript technically sound, and do the data support the conclusions?

Reviewer #2: Yes

3. Has the statistical analysis been performed appropriately and rigorously? 

Reviewer #2: Yes

4. Have the authors made all data underlying the findings in their manuscript fully available?

Reviewer #2: Yes

5. Is the manuscript presented in an intelligible fashion and written in standard English?

Reviewer #2: Yes

6. Review Comments to the Author

Reviewer #2: Thank you for addressing all my comments in detail. I appreciated the authors' comments and explanations regarding reverse-engineering; the text is now very descriptive and clear. I also noted that the authors have emphasized their reasoning for excluding the dropout layer both in terms of theoretical justification as well as providing experimental results. I now believe that their key contribution is better highlighted. The authors' comments on the quality of randomness of the input seed also helps in providing a better understanding of their implementation. The authors are encouraged to fix minor errors and typos in their manuscript (Eg. Pg 3 ln 81 "out"->"our", Pg 6 ln 204 "mode"->"model", Pg 7 ln 231 "cinsidered"-> "considered")

7. PLOS authors have the option to publish the peer review history of their article (what does this mean?). If published, this will include your full peer review and any attached files.

Reviewer #2: No

---

## [Editor Report · Acceptance letter]

5 Jun 2023

PONE-D-22-24211R1 

Learned Pseudo-Random Number Generator: WGAN-GP for Generating Statistically Robust Random Numbers 

Dear Dr. Kurabayashi:

I'm pleased to inform you that your manuscript has been deemed suitable for publication in PLOS ONE. Congratulations! Your manuscript is now with our production department. 

Kind regards, 

on behalf of

Dr. Sheetal Kalyani 

Academic Editor

PLOS ONE